# Characterization of Antibiotic Resistance and Metal Homeostasis Genes in Midwest USA Agricultural Sediments

**Michael R. Brooker [1],* , William A. Arnold [2] , Jill F. Kerrigan [2], Timothy M. LaPara [2], Jonathan D. Witter [3] and Paula J. Mouser [4]**

1    Department of Food, Agricultural, and Biological Engineering, The Ohio State University, Columbus, OH 43123, USA

2    Department of Civil, Environmental, and Geo- Engineering, University of Minnesota, Minneapolis-Saint Paul, MN 55455, USA; arnol032@umn.edu (W.A.A.); jllkrrgn@gmail.com (J.F.K.); lapar001@umn.edu (T.M.L.)

3    Agricultural and Engineering Technologies, Ohio State Agricultural Technical Institute, Wooster, OH 44691, USA; witter.7@osu.edu

4    Department of Civil and Environmental Engineering, University of New Hampshire, Durham, NH 03824, USA; paula.mouser@unh.edu

*    Correspondence: brooker.26@osu.edu

**Abstract:** Microbial antibiotic resistance is a naturally occurring phenomenon that has grown in part with the use of antibiotics in agricultural operation. There are also known connections between genes for metal homeostasis and antibiotic resistance, and either antibiotics or metals may select for both kinds of genes. Antibiotics, metals, and their associated genes have the potential to enter agricultural drainage channels and migrate to downstream locations through receiving water bodies. A relatively new agricultural best management practice—the two-stage channel design—functions by sequestering sediments and dissolved constituents as they flow through agricultural ditches from surface runoff and tile drainage discharge. Sedimentation in agricultural watersheds may entrap aggregate pollutants including antibiotics, metals, and associated resistance genes before transport into the drainage system. Here, we characterized the abundance and diversity of 22 antibiotic resistance and metal homeostasis genes in three two-stage channels that had self-developed in an area dominated by agricultural land use. Additionally, we analyzed the sediments for 17 antibiotics and nine metals that could affect the selection of these genes. In these rural systems that drain into Lake Erie, the abundance of antibiotic resistance and metal homeostasis genes were on the lower end of ranges (e.g., $<10^6$ gene copies g$^{-1}$ of *intI1*) reported in other riverine and lake systems, with only five genes—*intI1*, *aacA*, *mexB*, *cadA*, and *merA*—differing significantly between sites. The diversity of antibiotic resistance and metal homeostasis genes for these sediment samples were largely similar to other human impacted environments. Few antibiotics were detected in two stage channel sediments, with concentrations below the quantifiable limits (<0.02–34.5 µg kg$^{-1}$ soil) in most cases. Likewise, metals were present at what could be considered background concentrations. Despite serving as important drainage channel features in a region dominated by agricultural land use, results serve as an important baseline reference against which other monitoring studies can be compared to assess the perturbation of antibiotics and metals on agricultural channel sediments.

**Keywords:** agricultural drainage; two-stage channel; functional genomes; antibiotic resistance; metal homeostasis



## 1. Introduction

The transport of antibiotic-resistant microorganisms from the landscape into and through drainage channels poses a threat to downstream water supplies. Antibiotic resistant microorganisms and their genes (ARGs) can become enriched in soils due to management operations such as the application of manure (e.g., references [1–4]). ARGs have been detected in the runoff from edge-of-fields, in receiving water bodies, and in groundwater of agriculturally impacted areas [5–7]. Both antibiotic residues and ARGs are recognized as emerging contaminants [5], and their transport from agricultural sources needs to be closely monitored.

Antibiotic resistance by microorganisms is a naturally occurring phenomenon in soils, though it may be augmented by in situ selection—either by endogenous or exogenous antibiotics in the environment [8]. The spread of antibiotic resistance also occurs through horizontal gene transfer. For instance, ARGs have been associated with mobile genomic elements (e.g., integrons) that better allow these genes to spread across microbial community members [4,9–11]. Gene particles and antibiotics have been known to enter agricultural channels through surface and subsurface transport pathways [6]. Sediments carried into drainage channels may then act as reservoirs for antibiotic resistance microorganisms and/or genomic elements. Although ARGs have been well studied in other systems (e.g., wastewater facilities [12–15]), there is a lack of knowledge regarding the abundance and diversity of antibiotic resistance and the potential for ARG dissemination from agricultural sediments and their drainage systems.

In addition to selection by antibiotic residues, metals have been frequently noted as a co-selector of antibiotic resistance across a variety of environments [10,16–19]. This phenomenon may occur if the gene functions provide tolerance to additional stressors; or in cases where multiple genes are genetically linked on a common mobile genomic element (e.g., a plasmid or transposon) [16,19]. In the case of the former, multidrug efflux pumps have been proven to be capable of extruding heavy metals [10], while in the latter case, either metals or antibiotics could select for the shared genomic element [16]. Heavy metals are naturally present in most environments and some serve as macro/micro-nutrients. Yet, these metals can induce toxic and selective effects on the soil microorganisms if concentrations become too enriched [19]. Like with antibiotics, agricultural operations (e.g., fertilizer and manure application) lead to the enrichment of metals to the environment [17,18]. Thus, metals and metal homeostasis genes should also be considered when studying the spread of antibiotic in agricultural environments.

The Maumee River watershed in northwest Ohio is dominated (>75%) by agricultural land use [20], with modified drainage practices that include tile drainage systems and constructed, trapezoidal ditches [21,22]. Recent efforts to improve the natural functioning of drainage ditches includes the use of two-stage or self-forming channel designs [23]. Self-forming channels are created by widening the channeling and allowing sedimentation to naturally form floodplain benches [23]. The floodplains established within two-stage channels capture and remediate pollutants from both surface and tile drainage flows. This contrasts with riparian buffers implemented at the field elevations, which primarily treat surface runoff. Prior studies focusing on characterizing physicochemical processes and vegetation in restored two-stage channels show the design is effective at reducing dissolved nitrate, removing pesticides, and reducing the transport of sediments and specifically phosphorus, by increasing residence time through the floodplain [24,25]. Two-stage channels are likely to provide other ecosystem services, yet their role in mitigating organic and biotic contaminants has not been well studied. Specifically, two-stage channels may play an important role in sequestering antibiotics and ARGs, thus influencing the presence and diversity of antibiotic resistant microbial populations.

In order the characterize the abundance of antibiotic resistance and metal homeostasis genes in concert with antibiotic residues, we sampled three channels in the Maumee River watershed in which sediments had naturally deposited, as they would in a self-forming channel. We applied genomic tools (microarrays and quantitative polymerase chain reaction (qPCR)) to characterize the diversity and abundance of key functional genes. Analysis of select antibiotics and metals was further used to assess the abundance and diversity of constituents that may select or co-select for these genes.

Our objective was to quantify the abundance and characterize the diversity of antibiotic resistance and metal homeostasis genes in the sediment metagenomes to assess how agricultural activities may contribute to the spread of these genes through the drainage network.

## 2. Materials and Methods

### 2.1. Study Area and Sample Collection

Three locations were selected across the western Lake Erie basin (Figure 1). These sites were each located in a unique Level IV Ecoregion: Clayey, High-Lime Till Plains (CHLP), Oak Openings (OO), and Paulding Plains (PP) [26]. Criteria used in selecting these sites included (1) the presence of self-formed floodplains, (2) adjacency to agricultural row crop fields, and (3) greater than 70% agriculture land use in the watershed. Previously, our analysis of 15 floodplain sediments across five ecoregions revealed considerable variation in chemical and physical properties between the selected three sites (Appendix A). For example, our sites represented a gradient in soil textures (OO > CHLP > PP %Sand), total carbon and nitrogen (CHLP > OO > PP), while PP sediments had higher pH and the OO sediments had higher electrical conductivity (Figure A1).

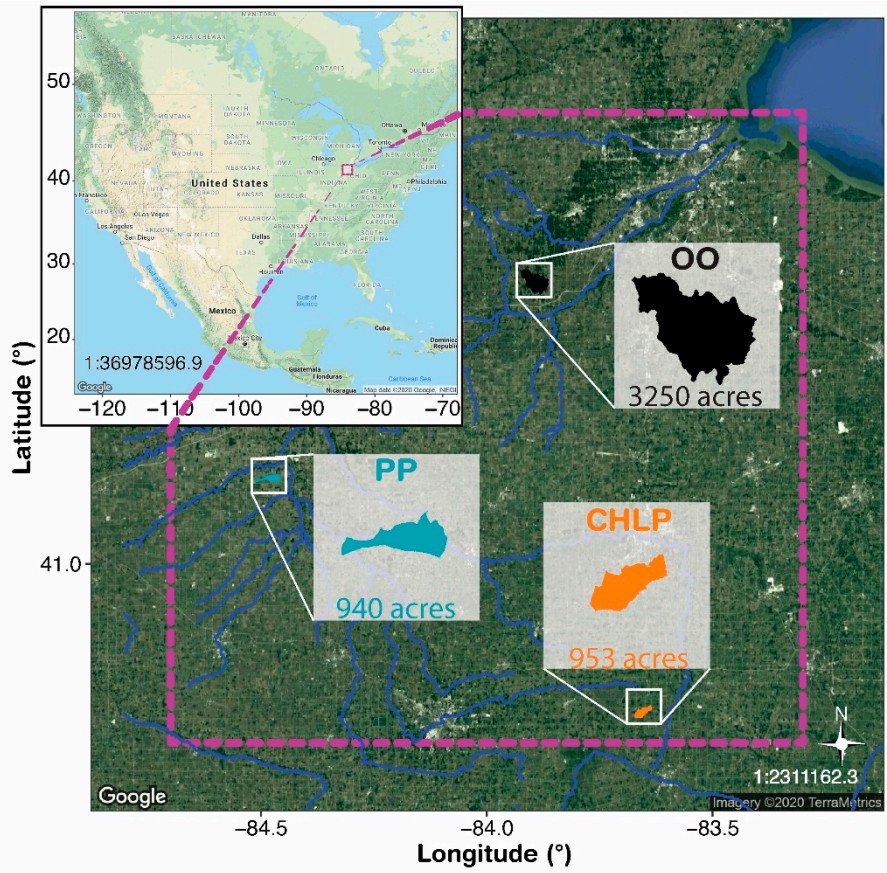

**Figure 1.** Map of the sampling locations used in the study. The study area was in northwest Ohio in the purple area highlighted on the map in the upper left corner (1:36978596.9 scale). Floodplain sediments were collected from benches formed in agricultural channels in the Oak Openings (OO), Paulding Plains (PP), and Clayey, High-Lime Till Plains (CHLP) Ecoregions as shown in the lower right map (1:2311162.3 scale). Watersheds were delineated using the US Geological Survey StreamStats (USGS) Geographic Information System (GIS) program and are shown for the sampling locations with the watershed area indicated.

Samples were collected on the same date in October 2016. Sediment cores were extracted from the surface of floodplains (0–20 cm) using a soggy bottom sampler device with sterilized PVC liners

(AMS Inc.; American Falls, ID, USA). The dry weights for sediments were calculated after drying more than 10 g of sediment at 70 °C until no further changes in weight were observed (approximately 24 h). For quantification of antibiotic concentrations, samples were collected in combusted glassware (450 °C, 2 h) using ceramic spoons. Samples were transported on ice to the laboratory where they were stored at −20 °C overnight. Samples used for antibiotic quantification were freeze-dried over the course of a week, while DNA extracts commenced on the day following sampling.

*2.2. Genomic DNA Extraction & Analyses*

Total nucleic acids were extracted using the MoBio PowerSoil DNA extraction kit (now Qiagen DNEasy PowerSoil Kit, Hilden, Germany) as previously described [27]. Triplicate cores were collected from each of the three ecoregions, homogenized, with 0.25-g aliquots used for DNA extraction. Replicate extracts were analyzed separately during Fluidigm qPCR (*n* = 9), while triplicate cores from each site were combined for GeoChip analysis (*n* = 3, see Supplementary Materials). DNA was stored at −80 °C until shipment on dry ice to sequencing and analysis facilities. Two platforms were used to analyze this DNA: (1) Fluidigm qPCR was used to quantify gene abundance; (2) the GeoChip 5.0 (Norman, OK, USA) was used to detect functional genes.

A combination of antibiotic resistance, metal homeostasis, and integrase genes were targeted for qPCR analysis using an integrated fluidic circuit (Fluidigm Corporation, San Francisco, CA, USA) [28] at the University of Minnesota. Briefly, 48 primer-sets and nine samples were input into a 48.48 Access Array (Table S1). EvaGreen dye was used at the fluorescent marker, which allowed for real-time quantification of amplification products. The amplicon pool was prepared by the Fluidigm FL1 and FL2 workflow. Several genes were targeted using multiple primer sets annotated with a gene suffix (e.g., *aadA* and *aadA*5) as described in Sandberg et al. [28]. Threshold cycle values were quality checked by Fluidigm software San Francisco, CA, USA). Standard curves for each gene were used to estimate the number of copies in each sample. The number of gene copies per gram of sediment was calculated based on volume of eluent used during DNA extraction and sediment mass extracted (Table S2). Detections outside the standard curve range or below detection limits were removed from the analysis.

The GeoChip 5.0 analysis was performed at Glomics, Inc. (Norman, OK, USA) as described by the manufacturer [29]. While qPCR was useful for quantifying the abundance of these genes, the GeoChip can detect the presence of species-specific variants of these functional genes, which is useful in describing the diversity of these communities. The GeoChip 5.0 microarray consists of 167,044 probes covering ~1500 functional gene families commonly observed in environmental systems. Briefly, purified DNA was labelled with Cy3 fluorescent dye with a random priming method and hybridized to the GeoChip 5.0 array slide. Slides were washed and scanned at 633 nm using a laser with a NimbleGen MS200 Microarray Scanner (Roche, Basel, Switzerland). Data was preprocessed by the microarray analysis by removal of poor-quality spots (SNR < 2.0). Each probe in the microarray targets a species-specific variant of a functional data. Metal homeostasis and antibiotic resistance genes detected in our samples are listed in the Supplementary Materials (Table S3). The data generated from this analysis was considered under a binary detect/non-detect basis. Raw and processed data for the GeoChip microarrays have been submitted to the NCBI GEO database and are publicly available under accession GSE125810.

*2.3. Analysis of Antibiotics and Metals*

Antibiotic extraction and quantification were carried out according to Kerrigan et al. [30]. Briefly, freeze-dried sediments were thawed and processed via accelerated solvent extraction with 50:50 methanol: 50 mM phosphate buffer (pH = 7). After removing the methanol via rotary evaporation, solid phase extraction (with Oasis HLB and MCX cartridges (Waters, Milford, MA, USA)) was used to concentrate and remove interferences prior to analysis. Liquid chromatography coupled with tandem mass spectrometry was used to detect and quantify the concentrations of 17 antibiotics commonly

used in agricultural operations. Full details of the method and quality assurance and control protocols are in Appendix B.

Metals analysis was conducted on air dried soils using microwave-assisted acid extraction (EPA 3051A) at the Service Testing and Research Laboratory (Wooster, OH, USA). Briefly, 0.5 g of the sediment was combined with 10 mL of 3:1 nitric to hydrochloric acid, microwaved to 175 °C, and held at that temperature for acid digestion over 10 min. Elemental analysis was conducted on purified digests using an Agilent 5110 inductively coupled plasma optical emission spectrometry (ICP-OES). This analysis measured the total concentration of a variety of elements associated with genes detected by the Fluidigm and/or GeoChip (Table S4).

### 2.4. Data Anlaysis

All analyses were completed using R Statistics (3.6.0) (Vienna, Austria) and its packages. Analysis of variance (ANOVA) was used to detect differences in the mean gene abundance ($log_{10}$), and multiple comparisons were made using the post-hoc Tukey HSD test. To compare the GeoChip results for our sediments to other environments, we downloaded the following sets from the NCBI GEO database: GSE67347, GSE92978, and GSE112489. These data represented 30 samples of restored grassland soils in Tibet [31], 48 samples of wastewater sludge [32], and 50 samples of pasture soils [33], respectively. Comparisons were first made between these studies using clustering analysis using the binary Jaccard (detected probes) approach in the "vegan" package [34]. For these genes, the unique probes associated solely with our two-stage channel sediments were identified using the "indicspecies" package using the *phi* coefficient corrected for an uneven number of samples [35]. Further, the Bonferroni method was used in determining statistically significant ($\alpha \leq 0.05/n$) probes. Images were plotted using the "ggplot2" package [36] with the integrated "ggwordcloud" package [37], while maps sourced from Google through the "ggmap" package [38].

## 3. Results and Discussion

### 3.1. Antibiotoic Resistance and Metal Homeostasis Genes Present at Natural Levels

The Fluidigm qPCR array was used to quantify the abundance of 46 antibiotic resistance and metal homeostasis genes in our sediment DNA. Of these 46 genes, 22 were successfully quantified (Figure 2). The most abundant gene quantified encoded the integrase of Class 1 integrons (*intI1*), a mobile genetic element commonly believed to aid in the proliferation of ARGs. This gene was present at more than $10^5$ genes per gram of sediment for each site. The abundance of genes varied ≤1-log across our set of samples. However, ANOVA testing followed by a post-hoc Tukey's test revealed 5 significant differences between our sediments ($\alpha < 0.05$). Both the PP and OO samples had a greater abundance of *intI1* gene copies. Otherwise, the PP sediments harbored a greater abundance of aacA and *cadA* genes while the OO sediments had a greater abundance of *merA* genes. The lowest abundance of metal homeostasis and antibiotic resistance genes were detected in the CHLP sediments. Yet, there were eight genes detected in the OO and CHLP samples that were not detected for the PP sample.

The abundance of *intI1* revealed an important concern. Integrons enhance the potential for recombination of ARGs between bacterial chromosomes and plasmids or other mobile genomic elements [39]. ARGs are known to be transferred either through conjugation or phage, or by carriage on plasmids with metal homeostasis genes [13,16,40]. For example, the abundance of *intI1* and ARGs were found to be correlated in the Minnesota and Mississippi Rivers [41]. Comparatively, the values reported here ($10^5$ gene copies) were at least one order of magnitude lower than those reported in the Mississippi and Minnesota Rivers ($10^6$–$10^7$) and soils amended with swine or dairy manure ($10^7$–$10^8$) [42,43], suggesting the impacts from the land use within the studied small, rural agricultural drainage systems was small by comparison. Interestingly, the values reported here for agricultural floodplain sediments were similar in magnitude to those in proximity to a WWTP outfall near Lake Superior ($10^5$) and other soils amended with wastewater sludge ($10^5$) [12,43] suggesting upstream

anthropogenic activity (including agricultural operations) strongly influences their abundance. Class I integrons have likely spread rapidly due to human activity—specifically in areas exposed to human and animal wastes and serve as an indicator for these sources of resistance [44]. The presence of these integrons suggests that human activity has influenced microbial communities to some degree in these floodplain sediments.

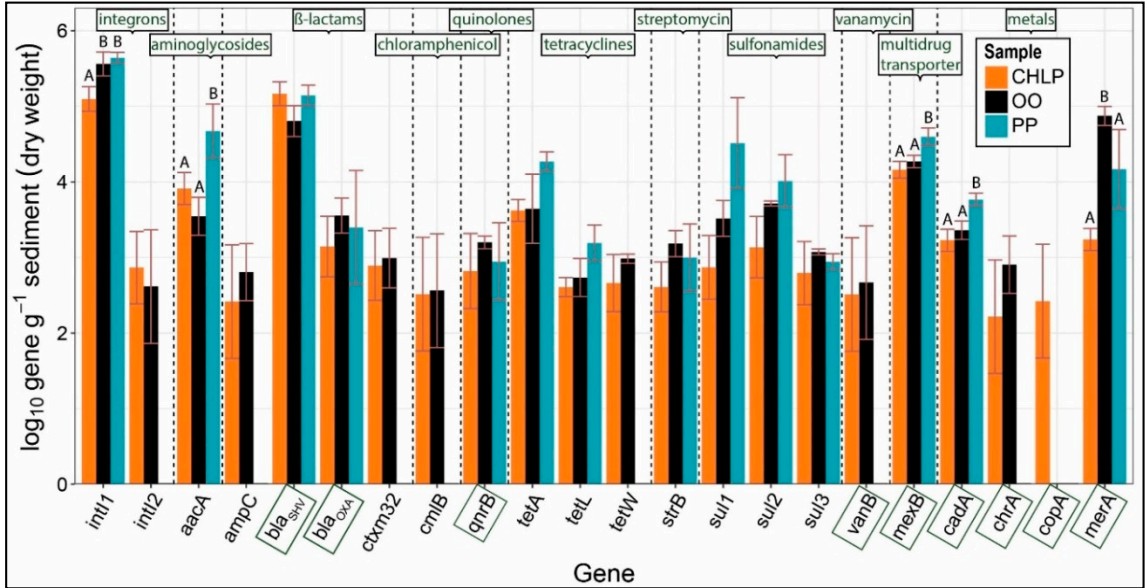

**Figure 2.** Fluidigm qPCR assay was used to quantify the gene abundance of a suite of metal homeostasis and antibiotic resistance genes with 22 gene successfully quantified. DNA extractions were performed on three separate cores collected at each site, with the standard deviation between these extractions illustrated by the error bars. Letters above the error bars indicate significant differences between the different sediments as calculated by the Tukey HSD test. Along the top of the graph, each gene is divided by dashed lines to the class of antibiotic to which they provide resistance. Genes which were also detected in the GeoChip are highlighted in boxes along the x-axis.

We used a complimentary molecular tool (GeoChip 5.0) to verify the presence, abundance and diversity of these genes. Nine of the genes targeted in the Fluidigm qPCR array were also included in the GeoChip microarray, although there were significantly more metal ($n = 115$) and antibiotic resistance genes ($n = 18$) detected by this device (Table S3). We therefore compared the pairwise functional diversity of shared antibiotic resistance and metal homeostasis genes across sites using the Jaccard dissimilarity index (Figure 3). The greatest dissimilarity was observed in the *copA* (copper transport); ß-lactamase class D (ß-lactam antibiotics hydrolysis, and includes the $bla_{OXA}$ gene from the Fluidigm,); and Van (e.g., *vanB*, D-alanine/D-lactate ligase of vancomycin) communities. On the other hand, there was less diversity between Mex (e.g., *mexB*; multidrug transporters) and *merA* (mercury detoxification) communities, and the ß-lactamase class A community (e.g., $bla_{SHV}$ gene in the Fluidigm) was the most similar across the three sediments. These results likely indicate that the environmental variations between our three sites had less of an influence on the development of Mex, *merA*, and ß-lactamase class A genes within the microbial communities.

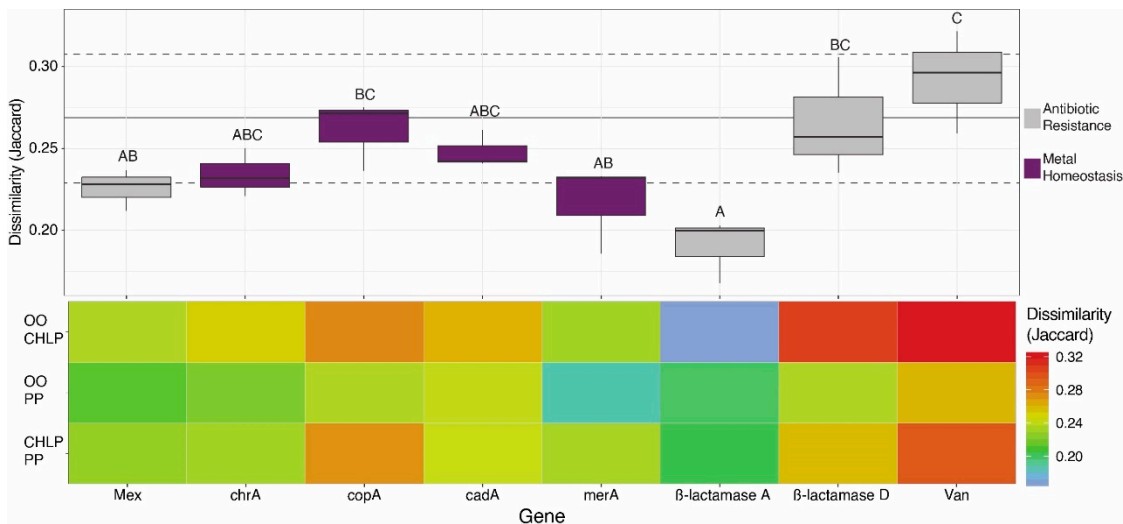

**Figure 3.** Diversity of antibiotic resistance and metal homeostasis genes between the sediments. Along the top are the box plot representing the three scores with letters indicating significant levels of difference between genes using the Tukey HSD test. The solid, horizontal line represents the average dissimilarity across the whole GeoChip bounded by the 95% confidence interval (dashed lines). The heatmap along the bottom illustrates the dissimilarity score calculated for each of the three sediment pairs (OO-CHLP, OO-PP, and PP-CHLP). Note: few *qnr* genes were detected by the GeoChip (only four across all sites); therefore, these genes were not included in the diversity analysis shown below (see Figure S1 in Supplementary Materials for additional details).

In addition to looking at overall diversity, we assessed pairwise dissimilarity between Ecoregion sites. Our sediments' genomes had remarkably similar functional diversity overall and across the antibiotic resistance and metal homeostasis genes in which the sites shared >70% of gene probes. As numerous other GeoChip studies have identified strong associations between microbial functional diversity and the physicochemical features of the system (e.g., references [29,45,46]), we expected the soil texture and geochemical variations between these three sites to relate to the similarity in antibiotic resistance and metal homeostasis gene diversity. Except for the highly similar ß-lactamase class A genes, we generally detected greater microbial community similarity between the PP and OO sediments as compared to the CHLP sediments. This was surprising considering their soil textures. The Paulding Plains (PP) and Clayey, High-Lime Till Plains (CHLP) are dominated by clayey soils whereas Oak Openings are primarily sandy (Figure A1) [26]. An alternative explanation is that PP and OO sediments had more similar carbon and nitrogen content, resulting in lower C/N content than that of the CHLP sediments. Both organic matter and nitrogen have been found to be significantly correlated with the functional gene diversity in rainforest and pasture soils [29,46]. Thus, antibiotic resistance and metal homeostasis diversity in these systems may be less affected by soil texture and more influenced by chemical features including carbon and nitrogen.

Each GeoChip is designed to detect a single gene-variant that arises from unique species providing the phylogenetic-lineage for each gene. Generally, the majority of probes detected for the antibiotic resistance and metal homeostasis genes were derived from *Proteobacteria* (Figure 4). In fact, >98% of all Mex genes—efflux-mediated resistance genes in the RND-superfamily of transporters—were derived from the Gram-negative *Proteobacteria.* This dominance of *Proteobacteria* was expected as RND transport systems span the inner and outer cell membranes of Gram-negative organisms [47]. Likewise, the presence of *Actinobacteria*-related Van genes makes logical sense as vancomycin affects Gram-positive bacteria through interfering with peptidoglycan development. For the metal homeostasis genes, *Proteobacteria* accounted for between 48–64% of the communities, while other abundant phyla included the *Actinobacteria*, *Firmicutes*, and *Bacteroidetes*.

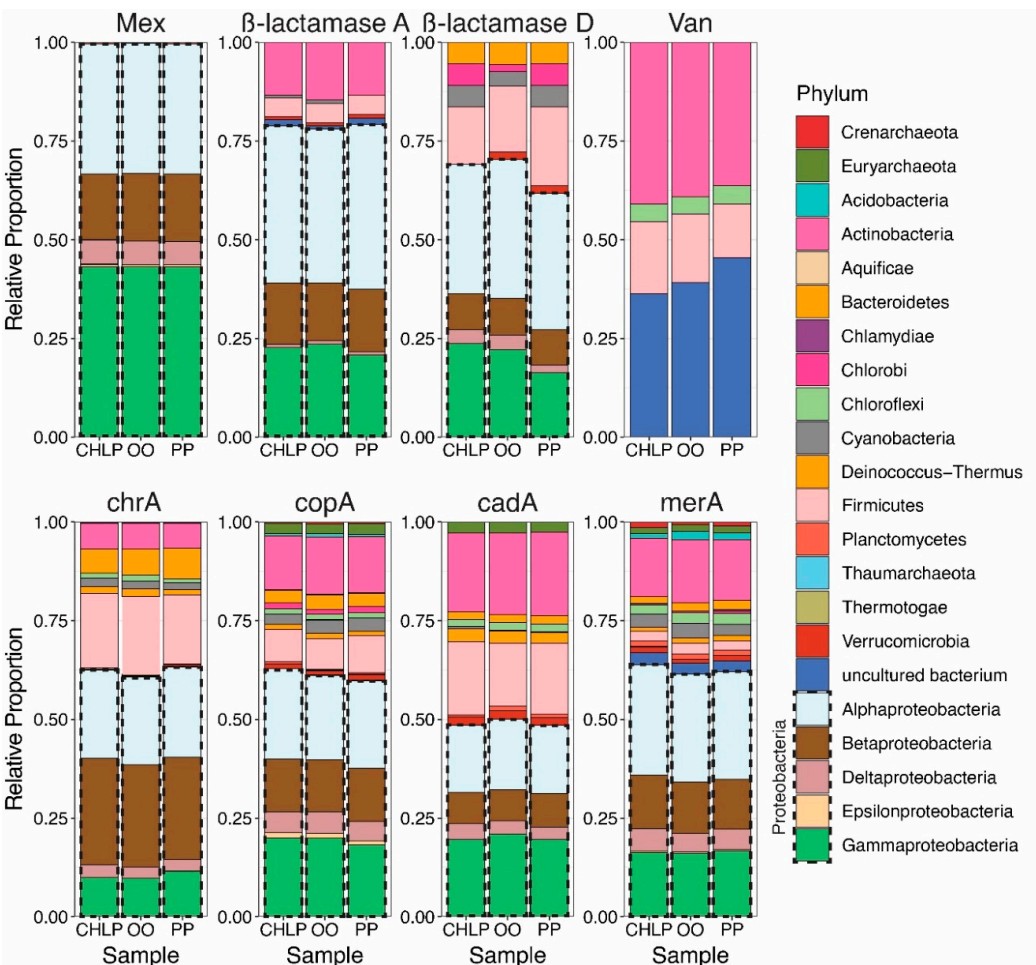

**Figure 4.** Relative proportion of the number of antibiotic resistance (**top row**) and metal homeostasis (**bottom row**) gene probes detected with lineages from designated phylum. The *Proteobacteria* are highlighted with the dashed lines but are further distinguished by their subphylum level.

The propensity for antibiotic resistance dissemination across community members must be recognized when it comes to this analysis. Horizontal gene transfer contributes to the rapid spread of ARGs across strains, species, or genera, although this can also occur across wider taxonomic (e.g., phyla) ranges [48]. Therefore, some of the genes targeted by GeoChip could be present in other, unrelated microorganisms than the gene lineages suggested. Furthermore, the integrons detected by Fluidigm provide an opportunity for ARGs to be disseminated across the community or may lead to future transfers [44]. While there are many nonpathogenic species of *Proteobacteria*, there are also many known pathogens [49] raising concern about the potential for horizontal gene transfer across this phylum in the drainage network. Whereas the efflux-mediated antibiotic resistance may provide resistance to multiple antibiotics [50], it is also important to note that this form of resistance has been known to aid in the regulation of metal toxicity [50,51]. Understanding the functional role played by these genes, and to confirm the presence of species hosting these genes would require the cultivation and isolation of the antibiotic resistance communities. This could also be used to screen for the presence of resistant organisms in these sediments.

To put these data into broader context, we compared GeoChip results for self-formed channel sediments to three other previously published data sets (Figure 5A and Figure S2 in Supplementary Materials). Across the eight selected antibiotic resistance and metal homeostasis genes, we found that the diversity in two stage channel sediments was largely similar to wastewater treatment plant sludge and pasture soil communities, sharing approximately 60% of gene probes. Notably, the wastewater

sludge, pasture soils, and two-stage channel sediments were all considerably different than those of restored Tibetan grassland soils, which were unimpacted by agronomic activities during the course of that study [31]. Two-stage channel sediments studied here formed a separate cluster, distinct from the wastewater sludge and pasture communities. As such, we were interested in determining which gene probes were uniquely associated with the two-stage channel sediments.

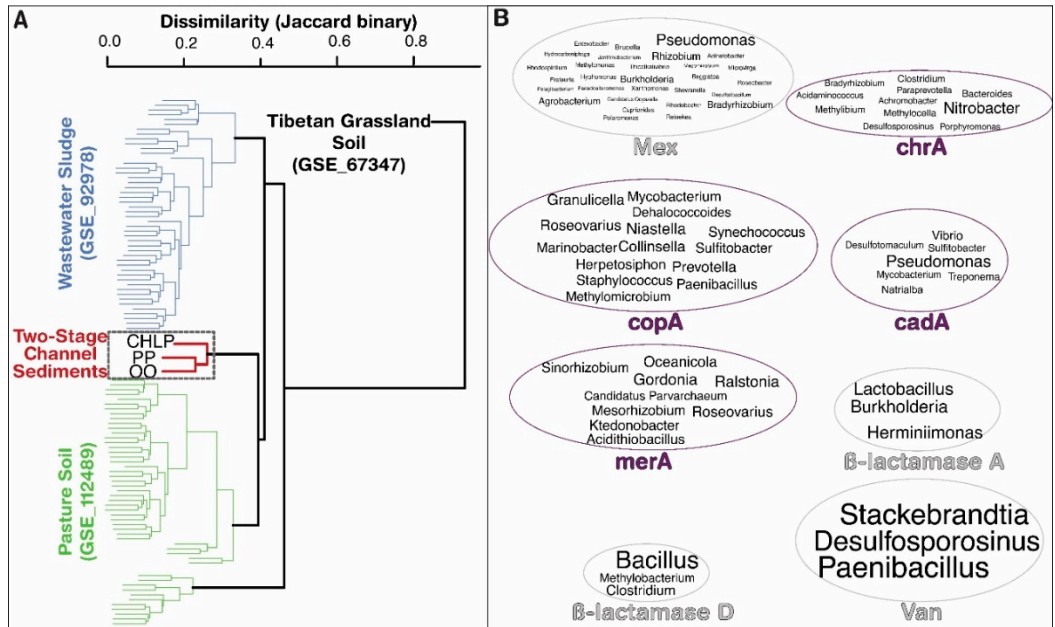

**Figure 5.** Antibiotic resistance and metal homeostasis gene probes were cross-detected in two-stage channels and other environments. (**A**) Hierarchal clustering analysis was used to compare different GeoChip sets based on the shared gene probes using the binary Jaccard index. Only one sample from a study of Tibetan restored grasslands is shown. (**B**) Indicator species analysis of antibiotic resistance and metal homeostasis gene probes for two-stage channel sediments with word area scaled according to the maximum number of these indicator gene probes associated with each genus and normalized to the number of probes for that particular gene (larger font = more gene probes).

Indicator species analysis of antibiotic resistance and metal homeostasis gene probes revealed that relatively few probes were uniquely associated with two-stage channel sediments (Figure 5B and Figure S2). The indicators solely assigned to the two-stage channel sediments often varied across the different genes. Only *Pseudomonas* (Mex and *cadA*); *Burkholderia* (Mex and ß-lactamase class A); *Bradyrhizobium* (Mex and *cadA*); and *Roseobacter* (Mex, *copA*, and *merA*) served as indicators of multiple genes. Thus, the distinction of two-stage channels arose from different species for each set of genes, rather than having these antibiotic resistance or metal homeostasis genes originate from common ancestors. For most of the genes, <2% of detected probes were solely identified as indicators of our two-stage channel sediments. The exception for this were the ß-lactamase class D and Van genes with both having the fewest number of probes detected across the GeoChip studies. There were many probes shared by pasture soils, wastewater sludge, and two-stage channel sediments that were not detected in restored Tibetan grasslands. Thus, it appears that the diversity of antibiotic resistance and metal homeostasis genes are relatively similar across these vastly different environments impacted by human activity.

### 3.2. Few Antibiotics, Natural Concentrations of Metals

To further understand the environmental conditions in which ARGs were detected, antibiotics were extracted from two sediment cores from each of our three sites (*n* = 6) and quantified using LC-MS/MS. Detection limits were determined for each antibiotic, ranging from 0.01–10.3 µg kg$^{-1}$

sediment. Only seven of 17 antibiotics were detected in at least one replicate, with 5 antibiotics detected in CHLP sediments, the most of any site sampled here (Table 1). Only one PP replicates tested positive for any antibiotic analyzed here (ofloxacin), while four antibiotics were detected in at least one replicate for OO sediments. Trimethoprim and erythromycin concentrations were above the limit of quantification in OO sediments (1.37 μg and 9.2 μg kg$^{-1}$ dry sediment, respectively) while other detections were below the detectable limit for both cases, suggesting antibiotics may be highly localized within these sediments.

**Table 1.** Concentration of antibiotics detected in agricultural floodplain sediment samples. Values below these detection limits are reported as <DL and italicized in gray font. Values below the limit of quantification are provided as estimates (est.). Readings that were quantifiable are bolded.

| Antibiotic | | Site | | | | | |
|---|---|---|---|---|---|---|---|
| | | CHLP | | OO | | PP | |
| | | Concnetration (μg kg$^{-1}$ Dry Sediment) | | | | | |
| Sulfanilamides | Sulfapyridine human | **<0.3** | est. 0.4 | **<0.4** | **<0.4** | <0.3 | <0.3 |
| | Sulfadiazine human, horse * | <0.02 | <0.02 | <0.02 | <0.02 | <0.02 | <0.02 |
| | Sulfamethoxazole human * | <0.3 | est. 0.5 | <0.4 | <0.4 | <0.4 | <0.4 |
| | Sulfamethazine swine, cattle * | <0.1 | est. 0.1 | <0.2 | <0.2 | <0.1 | <0.1 |
| | Sulfachloropyridazine swine, calf, dog * | <0.03 | <0.03 | <0.04 | <0.04 | <0.03 | <0.03 |
| | Sulfadimethoxine fish, poultry * | <0.1 | <0.1 | <0.3 | <0.3 | <0.2 | <0.2 |
| Uncategorized | Carbadox swine * | <0.06 | <0.06 | <0.06 | <0.06 | <0.07 | <0.07 |
| | Trimethoprim human, horse, dog * | <0.1 | <0.1 | <0.1 | **1.37** | <0.1 | <0.1 |
| | Lincomycin poultry, swine * | <0.01 | <0.01 | <0.01 | <0.01 | <0.8 | <0.8 |
| Tetracyclines | Tetracycline human, dog, cattle * | <0.04 | <0.04 | <0.02 | <0.02 | <0.02 | <0.02 |
| | Oxytetracycline fish, poultry, swine, cattle, sheep, bee, lobster * | <0.02 | <0.02 | <0.02 | <0.02 | <0.04 | <0.04 |
| | Chlortetracycline swine, poultry, cattle, sheep, duck * | <0.02 | <0.02 | <0.03 | <0.03 | <0.03 | <0.03 |
| Fluoroquinolones | Norfloxacin human, poultry * | <3.1 | <3.1 | <10.3 | <10.3 | <4.1 | <4.1 |
| | Ciprofloxacin human, poultry * | <1.3 | <1.3 | <4.5 | <4.5 | <1.7 | <1.7 |
| | Enrofloxacin swine, poultry, cattle, dog, cat | est. 0.8 | est. 1.0 | <0.8 | <0.8 | <0.7 | <0.7 |
| | Ofloxacin human, poultry * | est. 0.4 | est. 0.4 | <1.3 | <1.3 | <0.5 | est. 0.8 |
| Macrolides | Erythromycin human, poultry, swine * | <0.5 | est. 1.1 | **9.19** | <0.7 | <1.1 | <1.1 |

* Uses described by Meyer et al. [52].

Based on their known uses, antibiotics present in sediments help to distinguish between human and agricultural influence [52]. While we expected to find an influence of agricultural sources in the two-stage channel sediments, the concentrations of antibiotics were minimal, and detection was inconsistent across replicate cores. Thus, our study did not identify the influence of either human or agricultural inputs to the two-stage channels. This contrasts with the analysis of sediments from several Minnesota lakes where wastewater contamination was the likely source of most antibiotics [30]. Two-stage channel sediments had a greater resemblance to river sediments also collected from Minnesota, in which fewer antibiotics were detected [41]. Interestingly, we detected a larger number

of antibiotics in CHLP sediments, which was generally on the lower end of ARG abundances (10/18 genes, see Figure 2). Both sulfamethazine and enrofloxacin detected in the CHLP sediments and are not used by humans [52], but this detection was not reproducible. Notably, there were no known confined animal feeding operations within any of these three drainage areas, despite the predominance of agricultural land use surrounding the floodplain sampling sites [53]. This does not exclude nonpermitted animal operations or the import of manure fertilizer into these watersheds (~80% of manure in the watershed [53]) that may also introduce antibiotics to these watersheds. Overall, the studied two-stage channel sediments did not appear to accumulate antibiotics residues, though this does not preclude their usage or presence in upstream areas.

Contrary to antibiotics, a variety of metals were detected and quantified in our sediments (Table S4). Three of the four metal homeostasis genes detected by the Fluidigm had a corresponding metal concentration measured in this process: Cd (*cadA*), Cr (*chrA*), and Cu (*copA*) (Figure 6). While a greater abundance of *cadA* genes was detected in the PP sediments, the concentration of Cd in PP sediments was in between that of the CHLP and OO sediments. There was a greater concentration of Cr in the PP sediments, yet *chrA* was below detection. On the other hand, the CHLP had a greater concentration of Cu and was the only site where *copA* was also detected—but that occurred in only one of the three replicates. While significant differences in the quantities of *merA* genes had been noted, our methods did not allow for the detection of Hg in the sediments. In addition to these metals, a number of potentially toxic metals were quantified and corresponded with genes that had been detected in the GeoChip (Table S3), including Ni, Zn, As, Ag, Co, Pb, and Tl (Table S4).

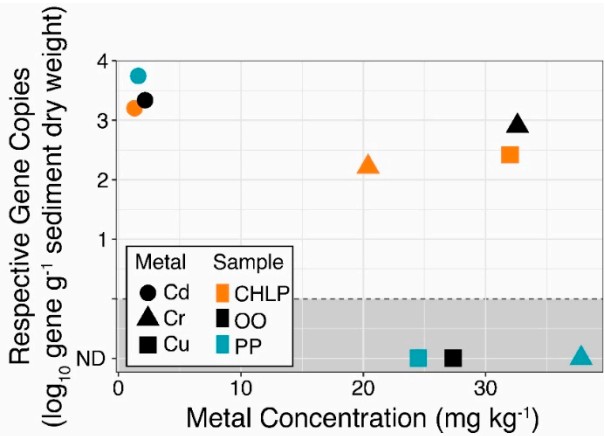

**Figure 6.** The concentrations of cadmium (Cd), chromium (Cr), and copper (Cu) in the three sediments compared to respective gene copies quantified by qPCR. Replicates were used to calculate the average metal concentration ($n = 2$) and gene abundances ($n = 3$). Note that mercury (Hg) was not quantified in this analysis. The dashed line represent $10^0$ and undetected (ND) genes are shown below this line in the gray, shaded area.

Although the metals are ubiquitously present in sediments, some may accumulate as a result of agricultural operations from pesticide and fertilizer application [54]. The concentrations of metals measured in Ohio floodplain sediment samples were generally within the range of those reported in Canadian and Chinese agricultural soils [54,55]. Cd alone had a mean concentration (2 mg kg$^{-1}$) considerably higher than sediments analyzed from Canadian (≤0.6 mg kg$^{-1}$) and Chinese (≤0.1 mg kg$^{-1}$) soils [54,55]. The concentration of Cd in the two-stage channel sediments even exceeded that of a stream fed by a tailing mine where a strong correlation was noted between this element and the abundance of its associated metal homeostasis genes. Interestingly, the *cadA* gene was one of only two metal homeostasis genes detected across all three sites. However, that mine tailing study found that Cd only correlated with the selection of *sul2* amongst ARGs [19], while there was no significant difference in the abundance of this gene across our two-stage channel sediments. The tailing mine drainage study

also showed that Pb correlated with the abundance of 10 ARGs, but their concentrations of Pb were over an order of magnitude higher than what was measured in our sediments [19].

*3.3. Future Research Needs for Agricultural Contributions to the Spread of Antibiotic Resistance*

While there were few differences and limited quantities of antibiotics, metals, and their associated genes in these sediments, the concern that agriculture contributes to the spread of antibiotic resistance genes remains well founded. Here, we sampled only from the sediments in naturally formed floodplains and did not capture the full spatial variation of these genes. Future studies should take a more holistic approach that includes collecting samples from agricultural fields, drainage water, and from the channel bed to get a better sense of antibiotic resistance occurrence. Further, wildlife in these settings may also act as vectors contributing to the occurrence of these ARGs [55,56]. Moreover, this study focused on only one aspect from the many ways in which antibiotic resistance may spread from agriculture. Considering other mechanisms for mobility would be beneficial.

Of the metals detected in these sediments, Cd seemed most likely to have a potentially toxic effect as this metal has a high toxic response factor and sediments with similar concentrations as reported here were associated with an increased abundance of metal homeostasis genes [19]. It will be important in future research to perform ecological risk assessments (e.g., Chen, J. et al. [19]) that can better identify the toxic thresholds to metal for sediment microorganisms. Additionally, evidence has shown that transport-mediated ARGs participate in the regulation of metal homeostasis [10], and so these types of genes should be given more focus in qPCR assays. While the *mexB* gene was quantified in abundance, one of the most common gene probes detected by the GeoChip—MFS multidrug transporters—were not included in the primer sets. Quantifying these genes could prove especially useful where metals are suspected of selecting for the spread of ARG from agricultural systems.

## 4. Conclusions

We characterized antibiotic resistance and metal homeostasis genes in sediment samples for three self-formed, two-stage channel sediments in Lake Erie (Ohio) tributaries. Metal homeostasis and antibiotic resistance genes are naturally present in aquatic environments, so their co-occurrence was expected. The data point toward limited influence of agricultural activities on genes encoding metal homeostasis and antibiotic resistance in two stage sediment samples. More likely, the three sites characterized here serve as useful background references in future monitoring studies. The two-stage channels had low levels of ARGs with a similar diversity of these and metal homeostasis genes as other human impacted environments. There were a limited number of unique probes associated with two-stage channel sediments which could distinguish these systems from other human impacted areas. While the two-stage channels we studied had a seemingly low abundance of ARGs and antibiotic residues, monitoring other channels in the area—particularly those adjacent to animal operations—could prove useful in determining how agronomic practices contribute to the environmental spread of antibiotic resistance, and whether two-stage channels can mitigate the migration of antibiotic resistance through drainage networks.

**Supplementary Materials:** The following are available online at http://www.mdpi.com/2073-4441/12/9/2476/s1, Figure S1. The number of gene probes detected in the GeoChip, Figure S2. The number of indicator species identified for each of the GeoChip studies, Table S1. Primers and standard sequences used for Fluidigm qPCR, Table S2. Yields and purity of DNA extracts of the sediments collected in October 2016. Table S3. List of all metal homeostasis and antibiotic resistance genes detected by the GeoChip 5.0, Table S4. Measured concentrations of elements in the sediments.

**Author Contributions:** Conceptualization, P.J.M. and J.D.W.; methodology, P.J.M. and M.R.B., formal analysis, M.R.B. and J.F.K.; investigation, M.R.B. and J.F.K.; resources, P.J.M., T.M.L., and W.A.A.; data curation, M.R.B.; writing—original draft preparation, M.R.B.; writing—review and editing, all authors; visualization, M.R.B.; supervision, P.J.M.; project administration, P.J.M. and J.D.W.; funding acquisition, J.D.W. and P.J.M. All authors have read and agreed to the published version of the manuscript.

**Funding:** This research was supported by a USDA National Institute of Food and Agriculture award to P.J.M. and J.D.W. (2012-51130-20255).

**Acknowledgments:** We are especially grateful to Andrew Ward for insight into the two-stage channel formation process for feedback in the development of project experimental design. We thank all the private landowners who granted access to their properties. We also thank Jeffrey Kast for checking for confined animal feeding facilities within our drainage watersheds; Julia Beni for running the Fluidigm qPCR analysis; and Glomics, Inc. for providing support with the GeoChip analysis. Thank you to the three anonymous reviewers for their valuable comments.

**Conflicts of Interest:** The authors declare no conflict of interest.

## Appendix A

Soil sampling was conducted in the spring (May–June) of 2013. At each site, the representative floodplain bench unit was divided into four quadrants: upstream, downstream, near bank, and near channel locations [57]. Each quadrant was sampled using an environmental sampling probe (2.6 cm diameter) lined with a clean plastic tube. Intact cores were collected primarily to determine soil physical properties, such as bulk density. Additional samples for nutrient and biological analyses were collected with a hand auger sampler marked at depths measured from the bench surface downwards. Samples were taken from discrete depth intervals at the surface (0–5 cm), followed by 15 cm increments (i.e., 5–20, 20–35, etc.) until reaching a depth corresponding to the channel bed ($n = 215$). Sediment samples were transferred to ice immediately following extraction and delivered within two days to the Soil and Bioenergy Lab at The Ohio State University South Center in Piketon, Ohio.

Samples collected with the environmental sampling probe were divided by depth, and 12-g subsamples were weighed, dried, and reweighed to calculate moisture content and bulk density. The remaining sample was combined with their respective hand-augured samples for use in quantifying chemical and biological properties. Soil (10 g) was mixed at a 1:1 ratio with DI water to determine electrical conductivity (EC) and pH with Thermo Fisher Scientific (Waltham, MA, USA) probes. The remaining sample was run on a Shimadzu TOC analyzer (Kyoto, Japan) to calculate labile carbon content. A 12-g (wet weight) sample was mixed with 30 mL of DI water for 2 h to collect the water-soluble reactive phosphorus (SRP) read using spectrophotometric methods [58] as orthophosphate. Samples were read spectrophotometrically on a Lachat continuous-flow analyzer (Milwaukee, WI, USA) at a wavelength of 882 nm.

Dried soil was weighed into 50-g samples that were exposed to hydrogen peroxide and microwaves to remove organic matter [59]. Remaining particles were placed in a 500 mL soil particle size analysis cylinder with 10 mL of 5% sodium hexametaphosphate. The sample was mixed, and a hydrometer was used to estimate particle size distribution by taking visibility measurements at 2 min (sand) and 2 h (clay). Silt was calculated as the portion that did not count as clay or sand. A Euclidean distance matrix was generated from the data and plotted using nonmetric dimensional scaling (NMDS). The factors of ecoregion, depth, and bench position were fitted to the ordination using the "vegan" package in R Statistics [34].

Initial screening of self-formed agricultural floodplain sediments in the Western Lake Erie Basin found the greatest variation in physical and chemical composition to occur across Ecoregions rather than by other spatial features such as depth (Figure A1). All sediments were primarily composed of sand (>50%), while silt composed less than 20% of all samples. Total carbon ranged between 11–124 mg kg$^{-1}$; total nitrogen ranged between 0.4–9.1 mg kg$^{-1}$; soluble reactive phosphorus (SRP) ranged between 0.01–5 mg kg$^{-1}$; and pH ranged between 7.2–8.6 across all samples. Visually, NMDS revealed the general trends observed for the soil properties across these different ecoregions.

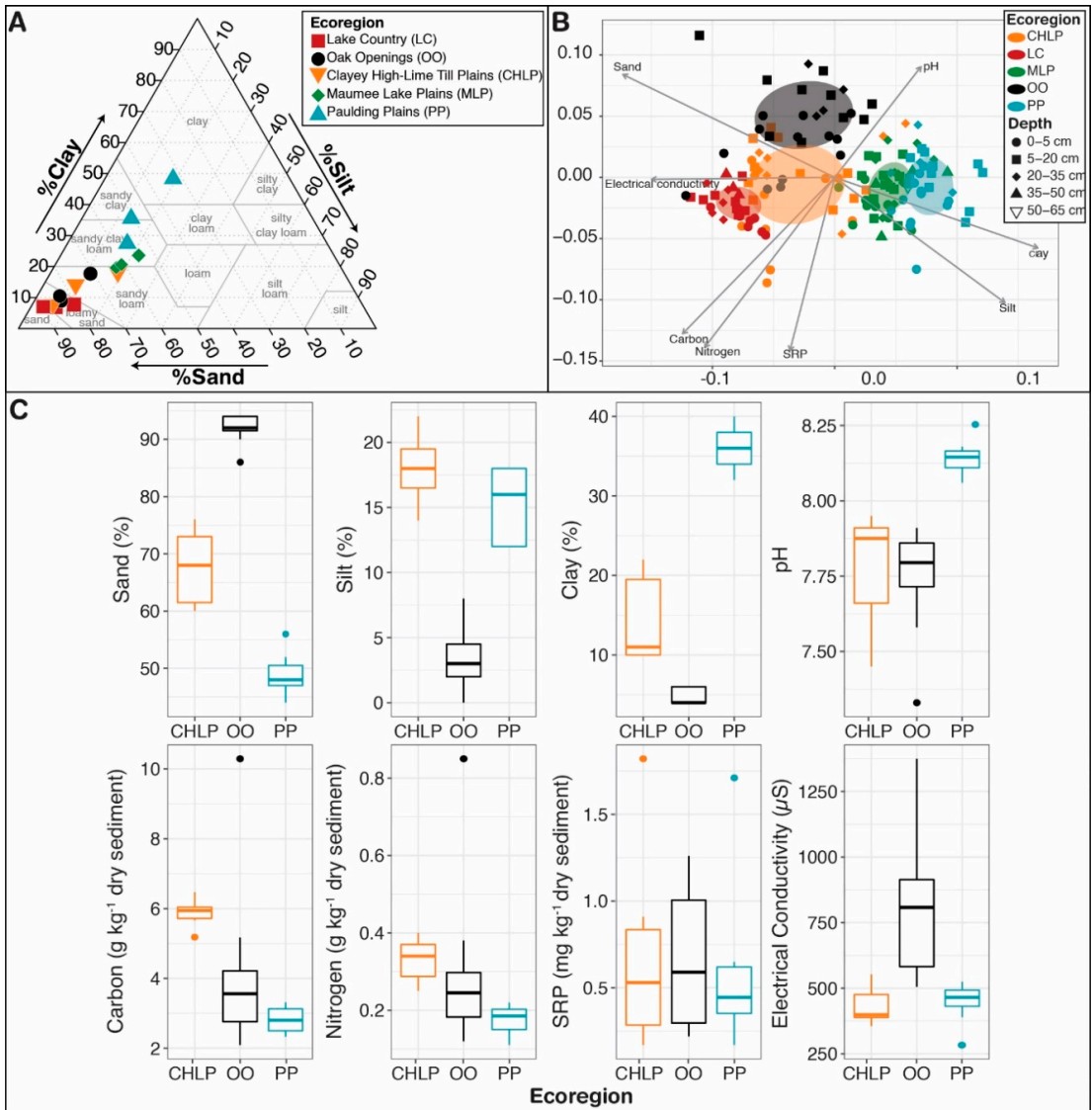

**Figure A1.** (**A**) Soil tests results for the three sites selected for further analysis for the top 20 cm (0–5 and 5–20 increments) from four bench positions. There are eight measurements made for each site with the boxes showing the 25th–50th–75th percentile (box), and whiskers showing the range of measurements. Outliers are shown as dots above or below the whiskers. (**B**) Nonmetric dimensional scaling (NMDS) analysis of the Spring 2014 data. Ecoregions (translucent circles) were significantly correlated to the ordination. The trends of the nutrients were plotted as arrows against this ordination. Each point represents the data from that sample and is colored by ecoregion and shaped by its depth. The plot was rotated to the principle coordinate axes (x = PC1). (**C**) Soil tests results for the three sites selected for further analysis for the top 20 cm from four bench positions (*n* = 24). Boxes cover the 25th–75th percentile with lines indicating median, and whiskers showing the range of measurements using dots for outliers (±1.5 × IQR).

## Appendix B

*Antibiotic Extraction and Quantification*

All glassware was triple-rinsed with a dilute Alconox solution, tap water, and DI water before being baked at 550 °C for more than 5 h to remove organic matter. Labware unable to be baked was triple-rinsed with acetonitrile, ethyl acetate, and methanol following the DI wash. Stainless steel accelerated solvent extraction cells were cleaned using the non-baking approach. Endcaps were rinsed

without the use of Alconox and then disassembled. The frit, cap insert, and snap fitting were soaked in a water bath and then sonicated in an acetone bath for 10 min. Following re-assembly, the organic solvent rinses were repeated.

Freeze-dried sediments were thawed and sieved prior to the extraction of antibiotics. Sediments (1 g) were spiked with surrogates (20 ng nalidixic acid and 100 ng $^{13}C_6$-sulfamethazine) in a methanol solution prior to accelerated solvent extraction (ASE) extraction. The ASE cells were assembled with two glass fiber filters, a thin layer of Ottawa sand, the sediment sample, filled with Ottawa sand, and covered with another glass fiber filter. A 50:50 methanol to 50 mM phosphate buffer (pH = 7) was applied at 100 °C for 5 min, allowed to sit for 5 min, with the process repeated twice and using a rinse volume of 150%. Methanol was removed from the ASE extract using a rotary evaporator in a 35 °C water bath.

Solid phase extraction (SPE) was adapted from Meyer et al. [60]. Oasis HLB (6 mL, 200 mg, 30 μm) and MCX (6 mL, 150 mg, 30 μm) columns were used in tandem, with the HLB column stacked on top of the MCX. Both columns were preconditioned with 10 mL of methanol and ultrapure water. Samples were loaded and passed through the column under a vacuum (<15 mm Hg). The HLB column was washed with 6 mL of 40:60 methanol:water, while MCX was washed with water (ultrapure). Antibiotics were eluted from the columns in tandem; first applying 3-mL of the extracts to the HLB column, and then applying 2 × 5 mL methanol to the MCX stacked on the HLB column. An addition elution of 3 mL 5% ammonium acetate in methanol was applied separately to the MCX column. The elution was initiated with a vacuum manifold but allowed to drip by gravity once started with the eluent collected in a 15-mL glass centrifuge tube. Internal standards (100 ng each of clinafloxacin, $^{13}C_2$-erythromycin, $^{13}C_2$-erythromycin-$H_2O$, simeton, and $^{13}C_6$-sulfomethoxazole) in methanol were spiked into the eluent. The eluents were dried under industrial grade $N_2$ in a 40 °C water bath. Samples were dissolved into 200 μL of 20 mM ammonium acetate, and syringe-filtered (GHP, 0.4 μm) to remove suspended particles prior to liquid chromatography tandem mass spectrometry analysis.

Samples were analyzed on a Thermo Fisher Scientific Dionex ultimate 3000 RSLCnano system (Waltham, MA, USA)equipped with a Thermo TSQ Vantage triple quadrupole tandem mass spectrometer (MS/MS) in positive electrospray ionization mode. Separation of antibiotics (8 μL injection volume) were achieved with a Waters XSelect™ CSH C18 (Milford, CT, USA) (3.5 μm, 130 Å, 50 × 2.1 mm) column at a flow rate of 0.5 mL/min and temperature of 35 °C. The elution buffer consisted of 0.1% formic acid in water or methanol and were applied at two gradients (Table A1). From 0 to 1.5 min and 5.5 to 20 min, flow was diverted to waste. Due to the number of analytes included in the study, each sample was analyzed by three LC-MS/MS methods that monitored for: (1) sulfonamides, $^{13}C_6$-sulfamethazine, and others; (2) tetracyclines, fluoroquinolones, and nalidixic acid; and (3) macrolides.

**Table A1.** Elution methods used in antibiotic quantification. Gradient elution of 0.1% formic acid in water and methanol (% B) with respect to time (min) that separated sulfonamides, macrolides, and others via method 1 and fluoroquinolones and tetracyclines via method 2 on a Waters XSelect CSH C18 column for ASE extracts.

| Method 1 | | Method 2 | |
|---|---|---|---|
| Time (min) | % B | Time (min) | % B |
| 0.0 | 0 | 0.0 | 0 |
| 5.5 | 100 | 0.5 | 0 |
| 7.5 | 100 | 4.0 | 40 |
| 8.0 | 0 | 7.0 | 100 |
| 20.0 | 0 | 9.0 | 100 |
| - | - | 10.0 | 0 |
| - | - | 20.0 | 0 |

Analytes were detected and quantified using single reaction monitoring (SRM) transitions (Table A2). Confirmation SRMs were used to corroborate the identity of quantified peaks. The mass spectrometer sensitivity varied between analyses, and thus, parameters were optimized with the infusion of 5 μM simeton in 50:50 20 mM ammonium acetate:methanol prior to each analysis. Typical values for mass spectrometer parameters were scan time 0.02 s; scan width: 0.15; $Q_1/Q_3$: 0.7; spray voltage: 3300 V; sheath gas pressure: 18 psi; capillary temperature: 300 °C; collision pressure: 1.5 mTorr; declustering voltage: −9 V; and tube lens: 95.

**Table A2.** Single reaction monitoring quantification and confirmation transitions and collision energy (CE) for analytes.

| Analyte | Parent Ion (*m/z*) | Product Ion (*m/z*) | CE (V) | Quantification or Confirmation |
|---|---|---|---|---|
| *Sulfonamides* | | | | |
| sulfapyridine | 250.10 | 156.00 | 17 | quantification |
| | 250.10 | 108.05 | 25 | Confirmation |
| sulfadiazine | 251.05 | 156.00 | 15 | quantification |
| | 251.05 | 108.05 | 24 | Confirmation |
| sulfamethoxazole | 254.05 | 92.10 | 29 | quantification |
| | 254.05 | 108.00 | 24 | Confirmation |
| sulfamethazine | 279.05 | 186.00 | 17 | quantification |
| | 279.05 | 156.00 | 20 | Confirmation |
| sulfachloropyridazine | 285.00 | 156.06 | 15 | quantification |
| | 285.00 | 92.05 | 35 | Confirmation |
| sulfadimethoxine | 311.10 | 156.06 | 21 | quantification |
| | 311.10 | 92.05 | 35 | Confirmation |
| $^{13}C_6$-sulfamethoxazole (*internal standard*) | 260.05 | 98.10 | 32 | quantification |
| | 260.05 | 114.10 | 27 | Confirmation |
| $^{13}C_6$-sulfamethazine (*surrogate*) | 285.05 | 186.00 | 22 | quantification |
| | 285.05 | 123.00 | 20 | Confirmation |
| *Fluoroquinolones* | | | | |
| norfloxacin | 320.10 | 276.10 | 17 | quantification |
| | 320.10 | 302.10 | 21 | Confirmation |
| ciprofloxacin | 332.10 | 231.05 | 35 | quantification |
| | 332.10 | 314.10 | 21 | Confirmation |
| enrofloxacin | 360.10 | 245.10 | 25 | quantification |
| | 360.10 | 316.15 | 19 | Confirmation |
| ofloxacin | 362.10 | 261.10 | 28 | quantification |
| | 362.10 | 318.10 | 19 | Confirmation |
| clinafloxacin (*internal standard*) | 366.10 | 348.00 | 20 | Confirmation |
| | 366.10 | 305.00 | 22 | quantification |
| nalidixic acid (*surrogate*) | 233.15 | 187.00 | 27 | Confirmation |
| | 233.15 | 104.05 | 40 | quantification |

**Table A2.** *Cont.*

| Analyte | Parent Ion (*m/z*) | Product Ion (*m/z*) | CE (V) | Quantification or Confirmation |
|---|---|---|---|---|
| *Tetracyclines* | | | | |
| Tetracycline | 445.10 | 410.10 | 19 | quantification |
| | 445.10 | 427.05 | 11 | confirmation |
| doxycycline | 445.10 | 321.05 | 31 | quantification |
| | 445.10 | 428.15 | 18 | confirmation |
| oxytetracycline | 461.10 | 426.10 | 17 | quantification |
| | 461.10 | 443.10 | 12 | confirmation |
| chlortetracycline & degradation products | 479.05 | 462.10 | 20 | quantification |
| | 479.05 | 444.10 | 17 | confirmation |
| | 481.05 | 464.10 | 20 | quantification |
| | 481.05 | 446.10 | 30 | confirmation |
| demeclocycline (*surrogate*) | 465.10 | 448.05 | 20 | quantification |
| | 465.10 | 430.05 | 17 | confirmation |
| *Macrolides* | | | | |
| erythromycin | 734.4 | 158.15 | 35 | quantification |
| | 734.4 | 576.35 | 15 | confirmation |
| erythromycin-$H_2O$ | 716.45 | 158.15 | 35 | quantification |
| | 716.45 | 558.35 | 15 | confirmation |
| roxithromycin | 837.45 | 158.10 | 35 | quantification |
| | 837.45 | 679.45 | 20 | confirmation |
| Tylosin | 916.45 | 174.10 | 40 | quantification |
| | 916.45 | 772.45 | 30 | confirmation |
| $^{13}C_2$-erythromycin | 736.40 | 160.15 | 35 | quantification |
| | 736.40 | 578.35 | 20 | confirmation |
| $^{13}C_2$-erythromycin-$H_2O$ | 718.40 | 160.15 | 35 | quantification |
| | 718.40 | 560.35 | 20 | confirmation |
| *Non-categorized* | | | | |
| Carbadox | 263.10 | 130.05 | 22 | quantification |
| | 263.10 | 231.05 | 13 | confirmation |
| Trimethoprim | 291.10 | 230.10 | 23 | quantification |
| | 291.10 | 123.05 | 24 | confirmation |
| Lincomycin | 407.30 | 126.10 | 35 | quantification |
| | 407.30 | 359.20 | 18 | confirmation |
| Simeton (*internal standard*) | 198.20 | 68.10 | 33 | quantification |
| | 198.20 | 100.10 | 27 | confirmation |

Several quality assurance and control measures were taken to assure the precision of reported antibiotic concentrations. Antibiotic extraction efficiency from sediment was determined for each collection site. This was achieved by spiking a methanolic solution of antibiotics (100 ng) onto the sediment prior to ASE and calculating the mass loss due to the extraction process. Method blanks (comprised of Ottawa sand spiked with surrogates) were subjected to the entire extraction process and were extracted at least every eight samples to monitor for any carryover contamination. Limits of quantification (LOQs) and detection (LODs) were defined as S/N ratio of 10 and 3, respectively. Antibiotic concentrations above LOQ were calculated using internal standard methodology and were corrected according to percent recovery. Reported LOQs and LODs were also corrected according to percent recovery.

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
