# Peer review of "Characterization of Antibiotic Resistance and Metal Homeostasis Genes in Midwest USA Agricultural Sediments"

_water, doi:10.3390/w12092476_

Round 1

Reviewer 1 Report

Title: Characterization of Antibiotic Resistance and Metal Homeostasis Genes in Midwest USA Agricultural Sediments (water-896837)

The manuscript prepared by Brooker et al., studied the abundance and diversity of antibiotic resistance and metal homeostasis genes in three two-stage channels that had self-developed in an area dominated by agricultural land use. It is important to understand the ARG and metal homeostasis genes together due to the exposure of antibiotics and metals. The manuscripts got some interesting results. However, there are some problems needed to be solved. Major revision is recommended.

Specific comments:

Line 30, five genes should be presented.

In the Introduction part, authors should add the relative researches about metal homeostasis gene, and how to connect with antibiotic resistance and metals.

Line 208-210, “While qPCR was useful for quantifying the abundance of these genes, the GeoChip…..the diversity of these communities”, this sentence is not relative the results and discussion, so should be removed to the Introduction part.

For Figure 6, y-axis should break the scale, because Zn concentration are relatively higher, thus causing the difference of metals with low concentration inconspicuous.

As reported that metals are not co-selecting for ARG and/or metal homeostasis genes in sediment microbial communities, so, how to explain the detected ARG and metal homeostasis genes together? In addition, authors should clarify the reason that how to select these kind genes.

Author Response

The manuscript prepared by Brooker et al., studied the abundance and diversity of antibiotic resistance and metal homeostasis genes in three two-stage channels that had self-developed in an area dominated by agricultural land use. It is important to understand the ARG and metal homeostasis genes together due to the exposure of antibiotics and metals. The manuscripts got some interesting results. However, there are some problems needed to be solved. Major revision is recommended.

Thank you for the interest in our research and for your comments which have helped to greatly improve this manuscript.

Specific comments:

Line 30, five genes should be presented.

We have listed the five genes in the abstract.

In the Introduction part, authors should add the relative researches about metal homeostasis gene, and how to connect with antibiotic resistance and metals.

We agree that this discussion was necessary and have added a paragraph going deeper into details about this relationship (Lines 62-73)

Line 208-210, “While qPCR was useful for quantifying the abundance of these genes, the GeoChip…..the diversity of these communities”, this sentence is not relative the results and discussion, so should be removed to the Introduction part.

We have moved this sentence to the Material and Methods (Lines 144-145) where qPCR and the GeoChip are first described.

For Figure 6, y-axis should break the scale, because Zn concentration are relatively higher, thus causing the difference of metals with low concentration inconspicuous.

In response to reviewer 3, we have revised this section. Figure 6 now shows the concentrations of the metals where qPCR detected/quantified a corresponding gene. The other metals previously included in this figure are now presented in Table S4 in the Supplementary Materials (Table S4).

As reported that metals are not co-selecting for ARG and/or metal homeostasis genes in sediment microbial communities, so, how to explain the detected ARG and metal homeostasis genes together? In addition, authors should clarify the reason that how to select these kind genes

We agree that this needed to be clarified. As ARGs and metal homeostasis genes co-occur naturally in aquatic environments, the presence of these genes should be expected. Lines 406-408 has been added to the conclusions to state this fact.

Reviewer 2 Report

This is a well written article with a really thorough methodology.

I only have 2 minor points:

-abstract: define lower end of range (line 29) and quantifiable limit (line 33)

-Information on the 48 primer sets sequences is missing. A supplementary table of primer sequences, or if not available, enough information to identify the manufacturers PCR primer mix used according to MIQE guidelines should be included

Author Response

This is a well written article with a really thorough methodology.

Thank you for the compliment!

I only have 2 minor points:

-abstract: define lower end of range (line 29) and quantifiable limit (line 33)

We thank this reviewer for this feedback. As there are multiple genes (22) and antibiotics (17) - each with their own limits – we did not include a comprehensive description. For the genes, we refer specifically to intI1 since this is a good indicator for anthropogenic impacts. On the quantifiable limit, we report the range of LoQs across antibiotics/sediments (Lines 31 and 35).

-Information on the 48 primer sets sequences is missing. A supplementary table of primer sequences, or if not available, enough information to identify the manufacturers PCR primer mix used according to MIQE guidelines should be included.

We thank the reviewer for noticing that information was missing. We have added Table S1 (cited at Line 135) containing this information to the supplementary materials.

Reviewer 3 Report

Manuscript by Michael R. Brooker, William A. Arnold, Jill F. Kerrigan, Timothy M. LaPara, Jon D. Witter and Paula J. Mouser

 Entitled: “Characterization of Antibiotic Resistance and Metal Homeostasis Genes in Midwest USA Agricultural Sediments”

 GENERAL COMMENTS

The study presents the results of an interesting research that aims to investigate the phenomenon of antibiotic resistance and that of metal contamination. It is necessary that the experimental plan would be presented in a more detailed and clear way to make the study analysis more robust.

Regarding metal contamination and antibiotic resistance, the authors are required to highlight the relationship between this two environmental contamination phenomenons.  

However, this relation is not explained in the manuscript while it should be highlighted starting from the abstract.

In particular, this must be reaffirmed in the introduction so that the aims of the work are consistent. A consideration that the authors should think about in the future concerns a comparison between the sediments of the surface of floodplains and those of the main channel of the same two-stage channel.

In addition, these aspects must also be analyzed in the discussion.

IN PARTICULAR:

 Abstract

- lines 24-26: Replace the sentence “Here we sought to characterize the abundance and diversity of antibiotic resistance and metal homeostasis genes in three two-stage channels that had self-developed in an area dominated by agricultural land use.” with “Here we characterized the abundance and diversity of 22 antibiotic resistance and metal homeostasis genes in three two-stage channels that had  self-developed in an area dominated by agricultural land use.”

Keywords

Add: “metal homeostasis”

  1. Introduction

- The part concerning the genes for the homeostasis of heavy metals should be expanded, this allows the reader to understand the importance of the proteins that manage the metal transport mechanisms contributing to homeostasis in organisms.

- In aquatic ecosystems the sediment matrix is an environmental reference matrix, however in the introduction section, that illustrates the various problems of the study, it is important to highlight and highlight the importance of bioassimilation through the use of bioindicators, i.e. the transfer of the actual contaminant (metal) from the environment to biota, see some references:

- Goretti E., Pallottini M., Cenci Goga B.T., Selvaggi R., Petroselli C., Vercillo F., Cappelletti D. (2018). Mustelids as bioindicators of the environmental contamination by heavy metals. Ecological Indicators, 94: 320-327.  https://doi.org/10.1016/j.ecolind.2018.07.004

-Alcorlo, P., Otero,M., Crehuet,M., Baltanás, A.,Montes, C., 2006. The use of the red swamp

crayfish (Procambarus clarkii, Girard) as indicator of the bioavailability of heavy

metals in environmental monitoring in the River Guadiamar (SW, Spain). Sci. Total

Environ. 366, 380–390.

- Also for the antibiotic resistance I would add that in the agricultural activity other matrices must be considered, as well as the sediments, whose contaminants can be easily detected through the use of bioindicators, see some references:

- Cenci-Goga B.T., Sechi P., Karama M., Ciavarella R., Pipistrelli M.V., Goretti E., Elia A.C., Gardi T., Pallottini M., Rossi R., Selvaggi R., Grispoldi L. (2020). Cross-sectional study to identify risk factors associated with the occurrence of antimicrobial resistance genes in honey bees Apis mellifera) in Umbria, Central Italy. Environmental Science and Pollution Research, 27: 9637-9645. https://doi.org/10.1007/s11356-020-07629-3

- Dolejska M. (2020) Antibiotic-Resistant Bacteria in Wildlife. In: The Handbook of Environmental Chemistry. Springer, Berlin, Heidelberg. https://doi.org/10.1007/698_2020_467

  1. Materials and Methods

- lines 94-96: the sentence “We expected that there would be a greater level of diversity between the functional genomes of our three sites based on these environmental factors.” is an anticipation of the results, remove it from Materials and Methods section

- lines 148-149: replace “Supplementary Materials (SM)” with “ Appendix B”

- lines 150-154:  add in detail in Appendix B (or in a new Appendix) all the detailed information on the analysis method for metals (wavelengths, detection limits, quality control, ...)

- add a new Appendix with a complete list of the 46 genes examined, so that the reader can know also the 24 not detected in the study

  1. Results and Discussion

- line 276: add a reference to figure S2, supplementary materials

- lines 311-312: this not seem to emerge from the table (not 4 but just 2)

- lines 335-336: were not 4 the genes found for 4 specific metals? Furthermore, the above metals are not consistent with figure 6 (there is no mercury, nor selenium, but there is thallium)

- reference 54 is too old for an adequate comparison with data from 1976

  1. Conclusions

Ok

 Appendix A

Ok

 Appendix B

Ok

References

See additions suggested above.

Author Response

GENERAL COMMENTS

The study presents the results of an interesting research that aims to investigate the phenomenon of antibiotic resistance and that of metal contamination. It is necessary that the experimental plan would be presented in a more detailed and clear way to make the study analysis more robust.

Regarding metal contamination and antibiotic resistance, the authors are required to highlight the relationship between this two environmental contamination phenomenons.  However, this relation is not explained in the manuscript while it should be highlighted starting from the abstract.

In particular, this must be reaffirmed in the introduction so that the aims of the work are consistent. A consideration that the authors should think about in the future concerns a comparison between the sediments of the surface of floodplains and those of the main channel of the same two-stage channel.

We thank the reviewer for the helpful comments and feel confident that the substantial revisions made should help to highlight the important relationship between metals and antibiotic resistance genes. We have added a line about the future research need for comparison between the floodplain sediments and channel beds – and agree that this area needs to be addressed as we could only find one paper that studied this variation. We hope that the changes in response to the comments below and from the other reviewers have helped to clarify the experimental plan.

In addition, these aspects must also be analyzed in the discussion.

IN PARTICULAR:

 Abstract

- lines 24-26: Replace the sentence “Here we sought to characterize the abundance and diversity of antibiotic resistance and metal homeostasis genes in three two-stage channels that had self-developed in an area dominated by agricultural land use.” with “Here we characterized the abundance and diversity of 22 antibiotic resistance and metal homeostasis genes in three two-stage channels that had  self-developed in an area dominated by agricultural land use.”

Thank you for this recommendation, we have made this change verbatim (Line 26-28)

Keywords

Add: “metal homeostasis”

We have added this as a keyword (Line 41).

Introduction

- The part concerning the genes for the homeostasis of heavy metals should be expanded, this allows the reader to understand the importance of the proteins that manage the metal transport mechanisms contributing to homeostasis in organisms.

We acknowledge that greater detail was needed about this relationship. We have revised the introduction to include a paragraph giving some background on this phenomenon (Lines 62-73). Metals and antibiotics primarily select for each other because either they (1) share a common mobile genomic element, like a plasmid or transposon, so the selection of one gene truly selects for both. The other possibility is that the function of one gene provides benefits to the organism for both stressors – the typical example is that a transporter extruding both metals and antibiotics.

- In aquatic ecosystems the sediment matrix is an environmental reference matrix, however in the introduction section, that illustrates the various problems of the study, it is important to highlight and highlight the importance of bioassimilation through the use of bioindicators, i.e. the transfer of the actual contaminant (metal) from the environment to biota, see some references:

- Goretti E., Pallottini M., Cenci Goga B.T., Selvaggi R., Petroselli C., Vercillo F., Cappelletti D. (2018). Mustelids as bioindicators of the environmental contamination by heavy metals. Ecological Indicators, 94: 320-327.  https://doi.org/10.1016/j.ecolind.2018.07.004

-Alcorlo, P., Otero,M., Crehuet,M., Baltanás, A.,Montes, C., 2006. The use of the red swamp crayfish (Procambarus clarkii, Girard) as indicator of the bioavailability of heavy metals in environmental monitoring in the River Guadiamar (SW, Spain). Sci. Total Environ. 366, 380–390.

Also for the antibiotic resistance I would add that in the agricultural activity other matrices must be considered, as well as the sediments, whose contaminants can be easily detected through the use of bioindicators, see some references:

- Cenci-Goga B.T., Sechi P., Karama M., Ciavarella R., Pipistrelli M.V., Goretti E., Elia A.C., Gardi T., Pallottini M., Rossi R., Selvaggi R., Grispoldi L. (2020). Cross-sectional study to identify risk factors associated with the occurrence of antimicrobial resistance genes in honey bees Apis mellifera) in Umbria, Central Italy. Environmental Science and Pollution Research, 27: 9637-9645. https://doi.org/10.1007/s11356-020-07629-3

- Dolejska M. (2020) Antibiotic-Resistant Bacteria in Wildlife. In: The Handbook of Environmental Chemistry. Springer, Berlin, Heidelberg. https://doi.org/10.1007/698_2020_467\

We have combined these two comments as they fell slightly outside the scope of the introduction section but were relevant to the greater picture of antibiotic resistance spread. We chose to focus this manuscript on the spread of ARGs through sedimentation in drainage channels. Likewise, we did not consider bioindicators of metals as there was insufficient funding of this research to allow us to seek insights into all of these aspects. As such, we have introduced a new Discussion section about future research directions that cites all of these references (Lines 385-403). We acknowledge there are more definitive ways to monitor for the spread of antibiotic resistance through the environment that would be necessary to state that the genes present in our sediments arose from sediment alone.

Materials and Methods

- lines 94-96: the sentence “We expected that there would be a greater level of diversity between the functional genomes of our three sites based on these environmental factors.” is an anticipation of the results, remove it from Materials and Methods section

This line has been removed from the Materials and Methods.

- lines 148-149: replace “Supplementary Materials (SM)” with “ Appendix B”

Thank you for bringing this to our attention, the appropriate change has been made (Lines164-165).

- lines 150-154:  add in detail in Appendix B (or in a new Appendix) all the detailed information on the analysis method for metals (wavelengths, detection limits, quality control, ...)

We have added this information along with the full analytical results to SI Table 4. We report the wavelength and instrument detection limits for all elements measured. Analytical details were provided by OSU OARDC STAR Laboratory: Samples were digested with quality control materials (Soil CK 13 [STAR lab] and NIST Standard 2709a). Additionally, blanks and standard check solutions were measured every 15-20 samples.

- add a new Appendix with a complete list of the 46 genes examined, so that the reader can know also the 24 not detected in the study

This request coincided with that of Reviewer 2: we have addressed this issue by adding a supplementary table containing the primers and standard sequences used in the microfluidic qPCR as Table S1 in the Supplementary Materials.

Results and Discussion

- line 276: add a reference to figure S2, supplementary materials

We have added this reference (Line 291).

- lines 311-312: this not seem to emerge from the table (not 4 but just 2)

We have corrected this to 6 antibiotics (sulfapyridine, sulfamethoxazole, sulfamethazine, enrofloxacin, ofloxacin, erythromycin) as this refers to detection in at least one of the two CHLP replicates. We have also ensured the text refers to detection in one of the two replicates as this has moved to Line 327.

- lines 335-336: were not 4 the genes found for 4 specific metals? Furthermore, the above metals are not consistent with figure 6 (there is no mercury, nor selenium, but there is thallium)

We thank the reviewer for noting these conflicting statements. We have revised Figure 6 to focus on the three metals included on the Fluidigm; and note mercury was not measured with our methods. The GeoChip contains a much wider variety of metal homeostasis genes which is why a selection of potentially toxic metals were included. We have added two SI Tables to support the revised paragraph. These include a description of all antibiotic resistance and metal homeostasis genes detected by the GeoChip (Table S3; Lines 226-227 and Line 363) and the full metal concentrations (Table S4; Lines 171-172, Lines 354-355). Therefore, the metals analysis continues to reference metal concentrations that may be relevant to toxicity, but this paragraph has undergone major revisions (Lines 371-384).

- reference 54 is too old for an adequate comparison with data from 1976

Please refer to the revision above, we have added more recent references regarding metal toxicity within the changes address in the previous comment.

Round 2

Reviewer 3 Report

The text revision greatly improved the manuscript. The new version, thanks also to the enrichment of the section of the Supplementary Material, has made the research results more understandable, giving a significant contribution to such important issues in an agricultural environment, as the relationship between the phenomenon of antibiotic resistance and the metal contamination.

However, the Authors have to make a final effort: double-check of the bibliographic citations in the text. Such a  tedious check is necessary, because it is essential for researchers interested in these investigations. 

In my opinion, after this amendment, together with the inclusion of  some suggestions, listed below,  the manuscript can be accepted for the final publication.

Additional  suggestions:

1) [55] Frank, R .; Ishida, K .; Suda, P. Metals in Agricultural Soils of Ontario. Can. J. Soil Sci. 1976, 56, 181-196.

This bibliographic citation is still in the “References” section even if it does not appear in the text of the manuscript

2) Lines 386-399: the bibliographic citations are reversed! Below it is the corrected version for bibliographic citations also considering the elimination of the citation [55]

- “3.3 Future Research Needs for Agricultural Contributions to the Spread of Antibiotic Resistance

While there were few differences and limited quantities of antibiotics, metals, and their  associated genes in these sediments, the concern that agriculture contributes to the spread of  antibiotic resistance genes remains well founded. Here, we sampled only from the sediments in  naturally formed floodplains and did not capture the full spatial variation of these genes. Future studies should take a more holistic approach that includes collecting samples from agricultural fields, drainage water, and from the channel bed to get a better sense of antibiotic resistance spread. Further, wildlife in these settings may also act as vectors contributing to the spread of these ARGs [55,56].”

  1. Cenci-Goga B.T., Sechi P., Karama M., Ciavarella R., Pipistrelli M.V., Goretti E., Elia A.C., Gardi T., Pallottini M., Rossi R., Selvaggi R., Grispoldi L. (2020). Cross-sectional study to identify risk factors associated with the occurrence of antimicrobial resistance genes in honey bees (Apis mellifera) in Umbria, Central Italy. Environmental Science and Pollution Research, 27: 9637-9645. https://doi.org/10.1007/s11356-699 020-07629-3
  2. Antibiotic-Resistant Bacteria in Wildlife. In: The Handbook of Environmental Chemistry. Springer, Berlin, 701 Heidelberg. https://doi.org/10.1007/698_2020_467

-“Moreover, this study focused on only one aspect from the many ways in which antibiotic resistance  may spread from agriculture. Considering other mechanisms for mobility would be beneficial.

Of the metals detected in these sediments, Cd seemed the most likely to have some toxic effect on the microbial communities indicated by relatively high concentrations and the detection of cadA in all three sediments. It is important to recognize that monitoring for the effects of metal toxicity  may be achieved using bioindicator communities [57,58].”

  1. Goretti E., Pallottini M., Cenci Goga B.T., Selvaggi R., Petroselli C., Vercillo F., Cappelletti D. (2018). Mustelids as bioindicators of the environmental contamination by heavy metals. Ecological Indicators, 94: 320-327. https://doi.org/10.1016/j.ecolind.2018.07.004.
  2. Alcorlo, P., Otero,M., Crehuet,M., Baltanás, A.,Montes, C., 2006. The use of the red swamp crayfish (Procambarus clarkii, Girard) as indicator of the bioavailability of heavy metals in environmental monitoring in the River Guadiamar (SW, Spain). Sci. Total Environ. 366, 380–390.

3) Bibliographic citations [60-62] are not reported in the manuscript text!

Author Response

The text revision greatly improved the manuscript. The new version, thanks also to the enrichment of the section of the Supplementary Material, has made the research results more understandable, giving a significant contribution to such important issues in an agricultural environment, as the relationship between the phenomenon of antibiotic resistance and the metal contamination.

However, the Authors have to make a final effort: double-check of the bibliographic citations in the text. Such a  tedious check is necessary, because it is essential for researchers interested in these investigations. 

In my opinion, after this amendment, together with the inclusion of  some suggestions, listed below,  the manuscript can be accepted for the final publication.

We sincerely apologize for the mistakes with the Bibliography and citations of these references. This error comes from not replacing the Frank et al. 1976 reference with a newer source on Canadian riparian sediments. Additionally, several citations were deleted but the references were left in the Bibliography. After careful review, all information in this section and all citations are associated with the correct reference. We are extremely grateful to the reviewer for noticing these errors! Please see our responses below

Additional  suggestions:

1) [55] Frank, R .; Ishida, K .; Suda, P. Metals in Agricultural Soils of Ontario. Can. J. Soil Sci. 1976, 56, 181-196.

This bibliographic citation is still in the “References” section even if it does not appear in the text of the manuscript

We have replaced this reference with:

  1. Saint-Laurent, D.; Hähni, M.; St-Laurent, J.; & Baril, F. Comparative Assessment of Soil Contamination by Lead and Heavy Metals in Riparian and Agricultural Areas (Southern Québec, Canada). Int J Environ Res Public Health. 2010, 7(8), 3100-3114

The statements made refer to the older work of Frank et al. were consistent with the findings of Saint-Laurent et al. and so there was only a minor modification to the text – that Cd in Canadian riparian sediments were <0.6 mg/kg instead of <0.7 mg/kg. We would like to reiterate our appreciation of this reviewer’s attention to detail in catching this error!

2) Lines 386-399: the bibliographic citations are reversed! Below it is the corrected version for bibliographic citations also considering the elimination of the citation [55]

Further, wildlife in these settings may also act as vectors contributing to the spread of these ARGs [55,56].”

  1. Cenci-Goga B.T., Sechi P., Karama M., Ciavarella R., Pipistrelli M.V., Goretti E., Elia A.C., Gardi T., Pallottini M., Rossi R., Selvaggi R., Grispoldi L. (2020). Cross-sectional study to identify risk factors associated with the occurrence of antimicrobial resistance genes in honey bees (Apis mellifera) in Umbria, Central Italy. Environmental Science and Pollution Research, 27: 9637-9645. https://doi.org/10.1007/s11356-699 020-07629-3
  2. Antibiotic-Resistant Bacteria in Wildlife. In: The Handbook of Environmental Chemistry. Springer, Berlin, 701 Heidelberg. https://doi.org/10.1007/698_2020_467

It is important to recognize that monitoring for the effects of metal toxicity  may be achieved using bioindicator communities [57,58].”

  1. Goretti E., Pallottini M., Cenci Goga B.T., Selvaggi R., Petroselli C., Vercillo F., Cappelletti D. (2018). Mustelids as bioindicators of the environmental contamination by heavy metals. Ecological Indicators, 94: 320-327. https://doi.org/10.1016/j.ecolind.2018.07.004.
  2. Alcorlo, P., Otero,M., Crehuet,M., Baltanás, A.,Montes, C., 2006. The use of the red swamp crayfish (Procambarus clarkii, Girard) as indicator of the bioavailability of heavy metals in environmental monitoring in the River Guadiamar (SW, Spain). Sci. Total Environ. 366, 380–390.

Again, we thank this reviewer for noticing these citation numbers were in reverse order. Following the exchange of references in the comments above these numbers are now:

  1. Saint-Laurent, D.; Hähni, M.; St-Laurent, J.; & Baril, F. Comparative Assessment of Soil Contamination by Lead and Heavy Metals in Riparian and Agricultural Areas (Southern Québec, Canada). Int J Environ Res Public Health. 2010, 7(8), 3100-3114
  2. Cenci-Goga B.T., Sechi P., Karama M., Ciavarella R., Pipistrelli M.V., Goretti E., Elia A.C., Gardi T., Pallottini M., Rossi R., Selvaggi R., Grispoldi L. (2020). Cross-sectional study to identify risk factors associated with the occurrence of antimicrobial resistance genes in honey bees Apis mellifera) in Umbria, Central Italy. Environmental Science and Pollution Research, 27: 9637-9645. https://doi.org/10.1007/s11356-020-07629-3
  3. Antibiotic-Resistant Bacteria in Wildlife. In: The Handbook of Environmental Chemistry. Springer, Berlin, Heidelberg. https://doi.org/10.1007/698_2020_467 Goretti E.;
  4. Pallottini M.; Cenci Goga B.T.; Selvaggi R.; Petroselli C.; Vercillo F.; Cappelletti D. Mustelids as bioindicators of the environmental contamination by heavy metals. Ecological Indicators, 94: 320-327. https://doi.org/10.1016/j.ecolind.2018.07.004.
  5. Alcorlo, P.; Otero, M.; Crehuet, M.; Baltanás, A.; Montes, C.; The use of the red swamp crayfish (Procambarus clarkii, Girard) as indicator of the bioavailability of heavy metals in environmental monitoring in the River Guadiamar (SW, Spain). Sci. Total Environ. 2006, 366, 380–390.

And the citations are now in the correct order:

While there were few differences and limited quantities of antibiotics, metals, and their associated genes in these sediments, the concern that agriculture contributes to the spread of antibiotic resistance genes remains well founded. Here, we sampled only from the sediments in naturally formed floodplains and did not capture the full spatial variation of these genes. Future studies should take a more holistic approach that includes collecting samples from agricultural fields, drainage water, and from the channel bed to get a better sense of antibiotic resistance spread. Further, wildlife in these settings may also act as vectors contributing to the spread of these ARGs [56,57]. Moreover, this study focused on only one aspect from the many ways in which antibiotic resistance may spread from agriculture. Considering other mechanisms for mobility would be beneficial.

                Of the metals detected in these sediments, Cd seemed the most likely to have some toxic effect on the microbial communities indicated by relatively high concentrations and the detection of cadA in all three sediments. It is important to recognize that monitoring for the effects of metal toxicity may be achieved using bioindicator communities [58,59]. Additionally, quantification of ARGs related to multidrug efflux should be given greater focus as these pumps have been known to expel heavy metals [10]. While the mexB gene was quantified in abundance, one of the most common gene probes detected by the GeoChip – MFS multidrug transporters – were not included in the primer sets. Quantifying these genes could prove especially useful where metals are suspected of selecting for the spread of ARG from agricultural systems.

3) Bibliographic citations [60-62] are not reported in the manuscript text!

We thank the reviewer for noticing that we should have removed these references following the modifications to the manuscript text. We now have the proper references corresponding to the citations in the Appendices.